# Barriers and enablers to access childhood cataract services across India. A qualitative study using the Theoretical Domains Framework (TDF) of behaviour change

Sheeladevi Sethu[1,2]◉*, John G. Lawrenson[1]◉, Ramesh Kekunnaya[3‡], Rahul Ali[2‡], Rishi R. Borah[2‡], Catherine Suttle[1]◉

**1** Division of Optometry and Visual Science, Centre for Applied Vision Research, City, University of London, London, United Kingdom, **2** Orbis International, Gurugram, India, **3** Child Sight Institute, Jasti V Ramanamma Children's Eye Care Centre, L V Prasad Eye Institute, Hyderabad, India

◉ These authors contributed equally to this work.
‡ These authors also contributed equally to this work.
* Sheeladevi.sethu@orbis.org

**Data Availability Statement:** All relevant data are within the paper and its Supporting information files.

## Abstract

Early presentation for childhood cataract surgery is an important first step in preventing related visual impairment and blindness. In the absence of neonatal eye screening programmes in developing countries, the early identification of childhood cataract remains a major challenge. The primary aim of this study was to identify potential barriers to accessing childhood cataract services from the perspective of parents and carers, as a critical step towards increasing the timely uptake of cataract surgery. In-depth interviews were conducted using a pre-designed topic guide developed for this study to seek the views of parents and carers in nine geographic locations across eight states in India regarding their perceived barriers and enablers to accessing childhood cataract services. A total of 35 in-depth interviews were conducted including 30 at the hospital premises and 5 in the participants' homes. All interviews were conducted in the local language and audio taped for further transcription and analysis. Data were organised using NVivo 11 and a thematic analysis was conducted utilising the Theoretical Domains Framework (TDF), an integrative framework of behavioural theories. The themes identified from interviews related to 11 out of 12 TDF domains. TDF domains associated with barriers included: 'Environmental context and resources', 'Beliefs about consequences' and 'Social influences'. Reported enablers were identified in three theoretical domains: 'Social influences', 'Beliefs about consequences' and 'Motivations and goals'. This comprehensive TDF approach enabled us to understand parents' perceived barriers and enablers to accessing childhood cataract services, which could be targeted in future interventions to improve timely uptake.

**Funding:** This study was funded by School of Health Sciences, City University of London as a research scholarship for the primary author's PhD. The funders had no role in study design, data collection and analysis, decision to publish, or preparation of the manuscript.

**Competing interests:** The authors have declared that no competing interests exist.

## Introduction

In cases of childhood cataract, early presentation for surgery is an essential first step in preventing associated visual impairment and blindness. The recommended age for congenital cataract surgery in children is within the first six to eight weeks after birth for unilateral cases [1, 2]. However, in India, cataract surgery in children is often delayed and a recent prospective study across India shows that the mean age at surgery for congenital and developmental cataract was 4 years and 8 years respectively [3]. Delayed cataract surgery in children has a profound effect on visual outcomes [4, 5]. In order to improve access, there is a need to understand the barriers and enablers associated with the access to childhood cataract services in India.

Access to eye care services for children depends on healthcare seeking behaviour (HCSB) of their parents and carers as well as the availability of eye care facilities in their community [6]. In this context, HCSB means the recognition of symptoms, timely presentation to health facilities and compliance with effective treatment. All these factors will influence postoperative visual prognosis.

HCSB is influenced by numerous factors including educational level, maternal occupation, marital status, economic status, age and sex, healthcare costs, women's status, type and severity of illness, distance and physical access, and perceived quality of service provision [7]. This wide range of factors indicates that the provision of education and knowledge at the individual level is not sufficient in itself to promote a change in behaviour [8]. Understanding HCSB more comprehensively at the level of the individual, the family and the larger community is likely to have benefits in addressing the gaps in service utilisation.

A number of models exist to explain healthcare seeking behaviour [9] but these are often not easily applicable by health professionals who do not have a psychology background [10].

The theoretical domains framework (TDF) was developed to make theories of behaviour and behaviour change more accessible to a range of users. The TDF has been cited in numerous peer reviewed publications focusing on behaviour patterns of health professionals and on patient uptake of healthcare services [11–13]. This framework has been used to analyse barriers and enablers in uptake of retinal screening among patients with diabetes [14, 15], and to understand professionals' perspectives on depression in a vision rehabilitation setting [16].

Behaviour change interventions are challenging and to achieve positive outcomes it is recommended that implementation strategies have a theoretical basis. Moreover, the interventions are more likely to be effective if they address the determinants (barriers and enablers) of the target behaviour [17].

It is important for programme planners to select a theory that is appropriate for the behavioural problem that they are trying to change with due consideration given to the setting/population. The Behaviour Change Wheel (BCW), is a framework that provides programme planners with a 'comprehensive, coherent, and universal toolkit for intervention design' and can be used to guide the choice of an appropriate intervention to achieve given outcomes [18].

This study uses the TDF approach and components of the behaviour change wheel to identify barriers and enablers associated with access to childhood cataract services in India. Based on these findings, various behavioural intervention functions are recommended.

## Methods

The consolidated criteria for reporting qualitative research (COREQ) guidelines were followed [19]. This study received approval from the School of Health Sciences Research Ethics Committee, City, University of London, and the institutional review boards of nine hospitals across India. Written informed consent was obtained from each participant individually.

## Study design

The primary intention was to understand the barriers which may in some cases be unique to the individual. In anticipation of barriers including financial and social factors which may be difficult for parents to discuss in a group in-depth individual interviews were conducted using a pre-designed topic guide developed for this study (S1 File). The topic guide was developed with the intention of helping the researcher to explore the barriers and enablers in depth and to stay focused on the topic and ensure all important questions were raised during the interview. However, the structure allowed the researcher sufficient flexibility to permit topics to be covered in an order most suited to the participants, to allow responses to be fully probed and explored and allow the researcher to be responsive to issues raised spontaneously by the participants [20].

## Participants and setting

A stratified purposive sampling technique was used to ensure that the participants included those at different socio-economic levels and with experience of accessing cataract services for their child in a range of hospitals across India. The details of the hospitals and locations has been reported elsewhere [3]. The stratification was conducted according to the child's surgery status including children who:

- had their cataract surgery without any delay, (defined as surgery completed within three months from recognition of the condition)

- had been advised to undergo surgery but had not done so within a 3-month period.

   Participants fulfilling the above two criteria were selected both from paying and non-paying patient categories at the participating hospitals. At each hospital location, 3 to 4 interviews were planned. Data saturation was reached when the data collection yields no new information about barriers and facilitators influencing access to cataract services. Saturation in interview studies is generally reached before 20 interviews. Hence, the target sample size for this study was 30 interviews with people whose children received cataract surgery [21].

   Additionally, 5 home interviews were conducted with parents and carers, whose children were advised to undergo cataract surgery but had not done so. For these home interviews, two geographic locations were selected using purposive sampling to obtain different perspectives from rural and semi urban areas. This cohort was identified from hospital records and for practical reasons the families residing within 15 kms of the hospital were selected. Telephone enquiries were made to check their availability and their consent to participate in the interview. Based on the verbal consent the home visits were made, and written consent was obtained from each participant (parent or carer) before the actual interview.

## Procedure

Data were collected from the participants between Nov 2015 and April 2016. All interviews were conducted in local languages (six different languages including Malayalam, Telegu, Hindi, Marathi, Bengali and Assamese) for the convenience of the participants to express their feelings openly during the discussion. Interviews were conducted in a separate room at the hospital to ensure privacy and to avoid any disturbances. Home interviews were conducted at pre-arranged appointments with the family according to their convenience.

   The interviews were audio taped with prior consent for transcription. The first author, who led the data collection and analysis, is female, was born and raised in India and had worked in the field of rural eye care for several years and thus was able to understand the local context

and cultural issues around it. The study was conducted as part of the author's PhD research, and her preparation for the qualitative data collection in this study was gained as part of PhD training and in her prior role in community eye health research. The author established no prior relationship with the study participants.

To minimise the possibility of the first author's ethno-racial background and cultural reflexivity influencing her interpretations of the data, a second author of United Kingdom origin also independently coded the transcripts.

All the hospital interviews and the home interviews were conducted by the principal researcher (SS) and a support staff member from the hospital. In most cases, the principal researcher was able to communicate with the participants in their first language. Only in two locations, Assamese and Bengali language interpreters with local community experience assisted with the interview. For each interview, the transcriber was also a translator if the interview was not in a language known to the principal researcher, providing an additional check of the interviewee's meaning. Each interview began with an introduction by the researcher and the parents were reassured that the interview was confidential and would not affect their child's treatment currently or in future.

## Data coding and analysis

A verbatim transcript of each interview was prepared from the tape recording of the sessions and imported into NVivo 11 for data management and analysis [22]. Transcription of all audio tapes was conducted externally by an independent company, and the researcher reviewed the transcripts for accuracy and completeness. The researcher was present in all the in-depth interviews, so it was possible to cross check with the field notes of each interview to ensure transcription accuracy. However, no major changes were made to transcripts based on the researcher's perception, retaining the participants' meaning. To maintain anonymity, participants' names were removed from transcripts. The researcher (SS) read all the 35 interview transcripts multiple times before coding the transcripts independently using the Theoretical Domain Framework (TDF).

## Development of coding framework

Quotes representing factors that helped or encouraged the parents/ carers to access the services early were coded as 'enablers' and the quotes that were associated with factors contributing to surgical delay were coded as 'barriers'. Every extracted statement was coded based on the 12 domains of TDF and its related component construct [10], either into one domain or into multiple domains. For example, one patient's father stated: "*We felt very sad. She is so young and she has got cataract*! *What would happen if we get her married*? *Problems can arise. So, without delay surgery should be done*". This was coded to both "Beliefs about consequences" ("what would happen...?") and "Social Influences" (related to marriage "Problems can arise...") domains.

Both deductive and inductive approaches were used to ensure all themes were coded [23]. As part of inductive analysis, the focus was on sifting and sorting the data to thematically synthesise and to identify key domains and key emerging issues under each identified domain. Two factors were used as "importance criteria" [15] to identify key domains which are likely to have the greatest influence on access to childhood cataract services.

1. The number of beliefs identified as barriers and enablers under each domain and elaboration (number of themes and sub themes) within each domain. The three domains with the

most frequently identified beliefs are discussed in detail in the results section. A summary of domains and their prioritization is provided in S1 Table.

2. The expressed importance within each domain. This was primarily a qualitative judgement made by the researcher based on her perception during each in-depth interview conducted in this research. For example, a point offered spontaneously by the interviewee rather than in response to a topic guide question was considered to have higher expressed importance.

A sample of 20% of all interview transcripts were randomly picked and coded independently by another researcher (CS) to increase the validity of the coding.

## Results

A total of 35 in-depth interviews were conducted including 30 at the hospital premises and 5 in the participants' homes. Time duration for interviews ranged from 40 to 90 minutes. The interviews were held with: i) both parents (n = 14), ii) mother (n = 10), iii) father (n = 6), iv) grandparents (n = 2) and v) family groups (n = 3) in which a few other relatives along with the parents also participated in the discussion.

Based on the interview transcripts, each quote was coded and themes representing barriers and/or enablers were identified. In total, the themes were consistent with 11 of the 12 TDF domains; no quotes were consistent with the domain of 'memory, attention and decision processes'. The S1 Fig shows the five most important (see Methods for importance criteria) TDF domains and the themes identified as barriers and enablers. Summaries of the number of quotes coded under each TDF domain as barriers and enablers with examples of quotes are shown in S2 and S3 Tables, respectively.

### Barriers to access of childhood cataract services

The three most important domains representing perceived barriers to childhood cataract services were found to be *environmental context and resources*, *beliefs about consequences* and *social influences*. Each of these are explained below, with examples.

**1. Environmental context and resources.** Most of the barriers identified by participants fitted this domain, and the most frequently cited barriers could be categorised under the following themes:

Economic reasons: The cost involved in seeking services was found to be the major barrier. Although the surgical services are available at no cost at the participating hospitals, the related costs such as travel, and lost wages may be significant. One participant stated: *I do labour job. . . It is labour job. If you go, you get (wages), if you don't go, you don't get. . .and I have two more children in the family. . . (Father, ref ID 16).*

Healthcare facility: Inadequate surgical eye care facility, lack of eye screening programs for children and the protocols and procedures followed at the hospitals were also identified as barriers within this domain. For example, the following comment reflects a lack of nearby surgical facility: *For eye problems people go to the doctors in nearby town. They would give medicines and a pair of spectacles. . . with spectacles one has to be happy. . . Other than that, there is no surgery service available close by. (Father, ref ID 4).*

Environmental stressors: These include issues related to distance to the hospital, transportation and the time taken for appointments. Most of the children's eye care facilities were located in larger towns and cities making it necessary for the families outside of these locations to travel long distances to access the services. Given the nature of surgical services and the number of follow ups required for the child pre- and post-operatively, parents and carers reported many challenges with various environmental stressors. An example comment from one child's

parents indicates difficulty reaching the healthcare facility for treatment: *They told [us] to come after 2 months. . . we have to see. . . there is no time, expenses too and we have to close the shop we run. . . it gets more difficult for other two daughters to go to school (Parents, ref ID 12).*

**2. Beliefs about consequences.** Participants expressed concern about perceived consequences as barriers that contributed to delay in accessing cataract services. The most important themes under this category included consequences, beliefs, and attitudes. Within the theme of *beliefs*, traditional / cultural beliefs based on spirituality and old practices were also found to be a major impediment for accessing cataract services for children. For example: *It happened since he was born. Now he is 4 years old. It is there since four years. We kept on thinking that it will go away, it will be cured. We kept on going to "babaji's place" [local priest]. People recommend this place to visit. . . to get cured. That's it . . . and the time kept passing on. . . (Father, Ref Id 13).*

Though some parents understand the consequences of delayed treatment, they may still delay accessing treatment due to a perception of other consequences such as negative outcomes post-surgery. One comment, for example, shows concern about harm related to the surgery: *If we tell him to bring something, he is unable to pick it up . . . I felt very sad but he is too small [young], so we didn't go for the surgery. If he rubs his eyes after surgery. . .he might get hurt himself. (Grandparents, ref Id 26).*

Lack of appreciation of the need for preventive care such as regular eye screening for early detection contributed to a delay in accessing cataract services in children, for example: *No. . .. we never thought the child should go for an eye check-up when there is no problem. . . Only if there is a problem, children should be taken for check-up. Otherwise not necessary. (Parents, Ref ID 10).*

**3. Social influences.** Social Influences also contributed significantly to the delay in accessing the services. The major themes identified as barriers under this domain were 'Social norms/ culture', 'Social Pressure' and 'Social support' at the family level.

Cultural and societal norms dictate to some extent the relatively low priority given to vision and eye care for children. Social and peer pressure for the parents to try alternative forms of treatment resulted in delay in accessing cataract surgical services in children. For example: *Yeah, people said about Ayurveda treatment [Herbal treatment]. We had tried with Ayurveda in parallel and came to know that by applying Ayurveda medicines in the eyes cataract could be cleared. When we came for the surgery we received calls suggesting us to not to go for the surgery. . . (Parents, ref ID 18).*

Pressure and a lack of empathy from peers acted as an emotional barrier among the children with cataract and their families. Parents in law played a major role in overall decision making for healthcare visits in several rural families. For example, under the domain 'Social norms/ culture' a mother in a rural area expressed this as: *I have accumulated the money for my son's operation . . . But, my in laws were not allowing us to go for surgery as everyone feels the child is too young. . . (Mother, ref ID 26).*

### Enablers to access of childhood cataract services

The three domains into which most of the enablers cited by participants fitted were 'Social influences', 'beliefs about consequences' and 'motivation, goals and intention'.

**1. Social influences.** The social structure of the family and the extended community had a significant impact on the decision-making processes around accessing cataract services. These included an individual's support in identifying or recognising the problem, help in arriving at an appropriate decision and most importantly the support extended either in accompanying the parent/carer to the hospital or looking after the young siblings when the parents are away

receiving hospital care. Those interpersonal processes that can cause individuals to change their views, feelings, or behaviours are grouped under this domain.

The major themes identified as enablers within this domain are as follows: "Social Influences" are 'Social support', 'Social norms/ culture' and 'change agents within the community'. For example: a parent in a rural area expressed this as:

*In the rural areas people don't have money. So, major decisions are taken after the family members sit together and decide on where the money will be arranged from and how. Sometimes neighbours and / or relatives are also involved. (Parents, ref Id 3).*

Many participants reported that deviations from social norms do occur. Some mothers had made the decision to take their child to the hospital when required even in rural communities where decision making of this kind is seen as a role for male members of the family. For example: *I take the decision as a mother and my husband doesn't say anything. Even in the neighbourhood as well men do not have time . . . men are busy with their work. Children's issues need to be looked after by their mothers. (Mother, ref Id 23).*

There were a few champions identified at the community level who are currently guiding the parents/ carers to seek appropriate care. They included the Accredited Social Health Activist workers appointed by the government to improve the health of rural communities. Significantly, parents whose children have already received care at the hospital were identified as champions of change within their communities.

**2. Beliefs about consequences.**   Parents who realised the importance of eye treatment for their child were more likely to take their child for treatment. The major themes identified as enablers under this domain are 'Attitude', 'Reinforcement' and 'Value'. For example: . . .*We felt. . . but, then we thought it is about the eyes and eyes are everything. So, surgery is necessary. My elder brother also came with me. So, we quickly took the decision to go ahead with the surgery and admitted her. (Parents, Id 10).*

The doctor—patient communication at the hospital played a pivotal role in enabling timely treatment, since they were in a position to reinforce the necessity of early treatment directly to the parents. For example: *The doctors told us that the surgery should be done immediately as with time the situation of the child would worsen. They showed me children aged as little as 4 months on whom surgery had been done and tried to convince me. They said that it is quite normal. After knowing all this, I became confident and went ahead with the surgery. (Father, Id 4).*

When the parent sees a benefit of treatment in their child, they are likely to give higher priority to spending time and money to make treatment possible. For example: *Doctor said that spectacle will be required for my child. So, I am getting it done here. There is no difficulty. . . Now my child is able to see. . . I will feel good to come back to the hospital. . .hoping that he would see better. (Father, ref Id 13).*

**3. Motivation, goals and intention.**   The themes covered under this domain include the motivation of the parents and their intentions towards seeking services for their children. When the parents had clear intention to provide early treatment for their child, they accessed the services early despite having economic challenges in the families. The major themes identified as enablers under this domain are 'certainty of the intention', 'Intention' and 'Intrinsic motivation / service intention'. For example: *Everyone has different thoughts. I think that even if I am doing labour job, my child should not do this. He should do some better job. So I would forego my wages to bring my child for check up to make sure his eye sight is good. (Father, ref Id 5).*

When the parents had positive intention towards the importance of eye sight for their child, they accessed the surgical services early. For example: *It is a matter of the eyes. Every person will think that his child should be alright. (Father, ref Id 28).*

Families whose children have gained vision after cataract surgery had clear intention to spread the benefits of early surgery in children among their communities. For example: *When we go back with our daughter, many people come to see. So, we will tell them that if your child has any eye problem, you should go to the nearest eye doctor. (Father, ref Id 16).*

There are other themes outside of the top 5 domains, but which may be important enabling factors for accessing childhood cataract services. The referral slip provided during the community screening program ensures confidence and motivation to access hospitals in different towns as parents were informed about the hospital location and where exactly they need to report within the hospital. For example: *. . . if had come alone, I would have been confused. . . what to do, where to go . . .when I came here, I had a referral slip with me from the community screening, where it was written 'contact counter no 4'. . .I went to the counter no 4, where the registration was done and then whatever was required at the hospital staff went ahead with their work and prepared the file. (Father, ref Id 5).*

## Discussion

This in-depth qualitative study is the first to explore the perceptions of parents and carers towards accessing childhood cataract services from multiple regions in India. This is also the first study to use TDF (which includes constructs from 33 behaviour change theories) in the field of community eye care to identify barriers and enablers for accessing childhood cataract services. The current study identified four TDF framework domains as the most influential factors as reported by parents and carers: 1) 'Environmental context and resources'; 2) 'Social influences'; 3) 'Motivations and goals'; 4) 'Beliefs about consequences'. Interventions that target these domains may be more likely to increase cataract surgery uptake in children.

A major concern reported by the parents and carers during the in-depth interviews related to the economic barriers to utilising the hospital services. Even when there is a possibility of free surgery at the hospital, the indirect costs associated with the treatment were reported to be a major barrier. Opportunity costs were a major issue as most of the parents had other family members including children and elders who depended on their income.

Previous research suggests that despite financial difficulties, healthcare utilisation is more likely if the illness is perceived to be either severe or life threatening [24]. In the case of cataract, parents and carers in the community were generally aware of the development and management of cataracts in adults. In particular, most adult cataracts tend to be treated in late adulthood and once the cataract is mature (described as 'ripe' in local terminology). In the present study, the elders in the family had significant involvement in decision making for health seeking behaviour. Those elders without the experience of childhood cataract may influence parents to delay surgery in children if they consider it benign and unlikely to have negative impact on vision.

However, economic issues did not deter parents from seeking surgery for their child if they were aware about the importance of early treatment and had access to a good healthcare facility. For example, parents with poor economic backgrounds expressed that, while they have no money available for cataract treatment, they were keen to raise money from other sources as they considered vision to be important for the better future of their child. This indicates the importance of creating awareness of the need for early treatment in this population. For example, the families need to be educated about the risk of losing vision permanently in children with cataract if the surgery is delayed.

## Implications for policy and practice

One of the key findings from this study is that establishing community eye screening programs in rural areas can act as both a barrier and an enabler for accessing cataract services in children (S2 and S3 Tables). The qualitative interviews revealed that some parents were waiting for the community screening program to be organised in their vicinity to access free services for their children, despite being aware of the problem and the need for early surgery. The inability to pay for the surgical services mean that they wait for the screening camp to arrive in the villages. However, in case of surgical referrals from the community screening, the parents reported confident in accessing the referred facility as they were provided with all the details in the referral form given to them.

Although most carers accessed eye care centres after recognising the eye problem, a substantial minority chose to access traditional healers and other forms of care initially, potentially delaying the opportunities for optimal intervention. Previous research in developing countries about health seeking behaviour for childhood illness [25] has indicated that care seeking behaviour in resource poor settings is a hierarchical process, in which carers first seek inexpensive solutions before visiting a hospital. Cultural beliefs based on religion and superstition were found to be an important impediment in accessing surgical services.

Another important factor that influenced access to childhood cataract services in this study was the social support provided by family and relatives and the wider community as a source of information and guidance. Cataract surgery requires admission for a minimum of three days at the hospital and mostly these hospitals are located in major towns and cities. If the family has more than one child, the support system within the family becomes an important enabling factor for cataract surgery. Arrangements need to be made for a member (usually a grandparent) to look after the other siblings whilst the parents are away. Similarly, after surgery a support network is required to ensure good post-operative recovery and later follow-up in children.

Previous work suggests that any decisions regarding the child's health and access to care are made at the household level and that these decisions are largely influenced by household factors such as parents' educational and occupational exposures and mainly depend on the household income [26]. A similar pattern was observed in the present research. Also, parents-in-law played a major role in overall decision making in rural families and this is consistent with previous findings [27].

The present findings also indicate that the eye care professionals in rural areas and in smaller towns may give inappropriate advice to the parents about cataract surgery in children (S2 and S3 Tables). This would have caused delayed presentation for cataract surgery and is modifiable with continuous medical education programs to update the local practitioners' knowledge in rural areas as they play a crucial role in timely referral to tertiary centres for surgical services.

Although Knowledge was not identified as one of the top three domains in this research, it is important to note that most of the participants were not aware of the specific issues regarding childhood cataract. However, the prior experience of the family related to adult cataract may have had a greater influence in the parent's health-seeking behaviour. This finding contrasts with earlier studies, in which knowledge among the parents about children's eye diseases was generally low but they were aware of cataract in children [28, 29]. This difference is due to the majority of the respondents in this study being from rural locations, where eye care service availability is very limited, whereas both previous studies were based on urban populations.

The findings of this research suggest that there are facilitators and barriers to childhood cataract services which are modifiable such as beliefs and consequences, social influences and

**Table 1. Recommended intervention functions for increasing the timely uptake of childhood cataract services in India.**

| Barrier domains* | Details of Barriers | Target audience | COM-B components** | Recommended intervention functions* |
|---|---|---|---|---|
| Beliefs about Consequences | A belief that it is acceptable to delay cataract surgery in children | Parents and carers | Reflective motivation | Education and Modelling |
| | A belief that a visit to babaji (local priest) will cure the cataract in children | | | |
| Motivation and goals | No intention to take the child for any routine eye examination, citing time constraints | Parents and carers | Reflective motivation | Education, Incentivisation, and Modelling |
| Environmental Context and Resources | Economic constraints and limited the feasibility of travelling long distances to seek the treatment | Parents and carers | Physical opportunity | Environmental restructuring, Training, and Enablement |
| Social Influences | Parents were influenced by what their friends and families did and recommended. | Parents and carers | Social opportunity | Environmental restructuring, training and Enablement |
| Knowledge | A lack of knowledge of cataract in children, and lack of awareness about the preventive aspects and when to go for surgery. | Parents and carers | Psychological capability | Education |

** COM-B component stands for Capability (Physical capability or Psychological capability), Opportunity (Physical opportunity or Social opportunity), and Motivation (Automatic motivation or Reflective motivation)–Behaviour, represents source of the behaviours and is the core of the BCW

* Recommended intervention functions were identified by the Behaviour Change Wheel (BCW)

knowledge, and this study is an important first step in establishing evidence as a basis for addressing issues on this topic. Further, this study has identified that there is a need to modify the health seeking behaviour of the parents and carers to address the issues related to childhood cataract services.

There are four steps involved in developing a theory informed implementaion intervention for achieving positive health seeking behaviour among the target population [17]. The first two of these involve identifying the specific behaviour and the target group and using the TDF approach to identify barriers and enablers that need to be addressed. Based on these two factors, the third step is to identify the intervention components that are feasible, relevant to the cultural context and defines the measures of behaviour change. This research has addressed the first two steps towards developing a theory information implementation. Based on the present finding this study has recommended various intervention functions using behavioural change wheel components of the COM-B model [18] to increase the timely uptake of cataract services in children Table 1. Programs implemented in wider communities should be developed that target specific behaviours that could bring changes in how childhood cataract and its treatment is perceived to increase the uptake of services by the communities. The impact of implementing such positive behavioural changes program in the community should be investigated using a robust experimental design.

## Strengths and weakness of the study

This is the first study to look systematically at the barriers and enablers for childhood cataract services in India using the TDF to identify theoretical perspectives associated with the identified issues. The respondents included in this research were selected from nine different geographical regions in India with broad cross-cultural representation. Also, the participants included parents, carers and other family members chosen from rural, urban and remote hilly and tribal areas to cover a range of perspectives among families in these regions.

The in-depth interviews were conducted with the parents and carers who had already accessed the hospital for childhood cataract surgery. The reported barriers and enablers from this study are therefore relevant to the members of the community who have accessed the services. Ideally, barrier assessment should target those who have been unable to access services

for their children's cataract, and further study including this group is required to identify barriers that have proved impenetrable for some.

In this study a purposeful sampling method was used. This method is widely used in qualitative research for the identification and selection of information-rich cases with relevant experience and with availability and willingness to participate, and the ability to reflect on and communicate their experiences for the most effective use of limited resources. However, the challenge in using this sampling strategy is that the range of variation in a sample is often not known at the beginning of a study or may be incorrectly assumed [30]. However, maximum variant samples were used to obtain varied experiences in utilising the services.

All parents or carers bringing their child to each of the included hospitals were eligible for inclusion. Selection bias was therefore not introduced at the point of recruitment. The hospitals were selected, and our results may have differed if a different set of hospitals had been included.

Most of the interviews were conducted at the hospital premises, either at the time of admission for surgery or during the follow up care. The hospital environment and the presence of a member of hospital support staff may have had an influence, for example parents and carers may have felt obliged to give favourable answers thinking that their response would affect their child's care. As part of the information provided to participants, they were assured that their participation in this research would not affect their child's treatment at the hospital. However, it remains possible that responses were tempered by the hospital location and the presence of a hospital staff member, and that the use of a more neutral location without hospital staff may yield more barriers and enablers.

Another important limitation in this study is the involvement of language interpreters to facilitate the discussion in few locations. This has an inherent disadvantage on the flow and continuity of the discussion and may potentially impact on the interaction with the participants. The presence of an interpreters may have made participants less willing to voice any concerns. In addition, it is possible that the translation itself may have altered the meaning or emphasis in the participant's words, and thus may have affected the results. However, the person engaged for transcribing the interviews, who was unrelated to this research were able to capture the exact information conveyed by the participants.

Data was initially analysed deductively, using the TDF to generate the framework for an inductive analysis that generated themes within TDF domains. 'Whilst the TDF provides a useful and comprehensive theoretical approach to identifying influences on behaviour, if applied too rigidly there is a risk that non TDF-related participant perspectives could be missed. We attempted to reduce this risk by applying a more flexible inductive analytical approach to ensure that potential themes that could not be coded to the TDF were not lost.

## Future research

Childhood cataract surgery requires long term follow up and achieving a good visual outcome after cataract surgery is likely to depend at least in part on the post-operative follow up care in children. Achieving maximum uptake of follow up care is a continuous challenge even in adults for any chronic conditions and it is considered more challenging in children. The present research was focused on access to services and additional research is needed to assess the barriers and enablers associated with the access to follow-up care post childhood cataract surgery in different regions.

## Conclusion

Our findings highlight that the TDF is considered as a useful approach providing a systematic, comprehensive, and theory-derived process to identify barriers to access childhood cataract

services that can help identify target behaviours for change and inform implementation strategies. Also, this study found that the TDF was a flexible approach that could be used across different settings and in different ways to understand planning and implementation of relevant activities.

## Supporting information

**S1 File. Topic guide for in-depth interviews with parents.**
(DOCX)

**S1 Table. Summary of TDF domains for barriers and enablers.**
(DOCX)

**S2 Table. Summary of statements classified as barriers, sorted by TDF domain.**
(DOCX)

**S3 Table. Summary of statements classified as enablers, sorted by TDF domain.**
(DOCX)

**S1 Fig. Top 5 TDF domains identified as barriers and enablers in accessing childhood cataract services.**
(TIF)

## Acknowledgments

The authors acknowledge the support of staff and management of all participant hospitals, parents and children who have contributed to this study.

## Author Contributions

**Conceptualization:** Sheeladevi Sethu.

**Data curation:** Sheeladevi Sethu.

**Formal analysis:** Sheeladevi Sethu.

**Investigation:** Sheeladevi Sethu.

**Methodology:** Sheeladevi Sethu, John G. Lawrenson, Ramesh Kekunnaya, Rahul Ali, Rishi R. Borah, Catherine Suttle.

**Project administration:** Sheeladevi Sethu.

**Supervision:** John G. Lawrenson, Catherine Suttle.

**Validation:** Catherine Suttle.

**Writing – original draft:** Sheeladevi Sethu.

**Writing – review & editing:** Sheeladevi Sethu, John G. Lawrenson, Ramesh Kekunnaya, Rahul Ali, Rishi R. Borah, Catherine Suttle.

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
