## [Decision Letter · Decision Letter 0]

16 Sep 2021

PONE-D-21-05307Barriers and enablers to access childhood cataract services across India. A qualitative study using the Theoretical Domains Framework (TDF) of behaviour change

PLOS ONE

Dear Dr. Sethu,

Thank you for submitting your manuscript to PLOS ONE. After careful consideration, we feel that it has merit but does not fully meet PLOS ONE’s publication criteria as it currently stands. Therefore, we invite you to submit a revised version of the manuscript that addresses the points raised during the review process.

We look forward to receiving your revised manuscript.

Kind regards,

Leeberk Raja Inbaraj, MD

Academic Editor

PLOS ONE

Journal Requirements:

2. Please ensure that all references are cited correctly. For example, we note that in the Methods, the sentence " The details of the hospitals and locations has been reported elsewhere (1)" seems to refer to the wrong citation.

3. Thank you for stating the following financial disclosure: "NO"

4. Thank you for stating the following in your Competing Interests section: "No authors have competing interests"

Reviewers' comments:

Reviewer's Responses to Questions

**Comments to the Author**

1. Is the manuscript technically sound, and do the data support the conclusions?

Reviewer #1: Yes

Reviewer #2: Yes

2. Has the statistical analysis been performed appropriately and rigorously? 

Reviewer #1: Yes

Reviewer #2: N/A

3. Have the authors made all data underlying the findings in their manuscript fully available?

Reviewer #1: Yes

Reviewer #2: No

4. Is the manuscript presented in an intelligible fashion and written in standard English?

Reviewer #1: Yes

Reviewer #2: Yes

5. Review Comments to the Author

Reviewer #1: Thank you for affording me the opportunity to review this study. This study aimed to identify potential barriers to accessing childhood cataract services from the perspective of parents and carers as a critical step towards increasing the timely uptake of cataract surgery in India. The study presents an important topic and is timely. It further presents constructs from behaviour change theories that could be applied in other public health areas. I therefore think that the work is publishable but there has to be minor revisions before then.

General:

- There needs to be consistency in writing “healthcare” and “health care”, choose one. My personal preference if the former.

Introduction:

- Line 3 of Introduction: “…is congenital cataract…” should be removed as it's a typo.

- Last paragraph of introduction has a typo in the first sentence second line of that paragraph. The word “…the…” should be removed (i.e ‘…the access to childhood cataract services…’ should read as ‘…access to childhood cataract services…’.

Methods:

- Considering that this is a qualitative study the researchers should present statements on reflexivity and positionality of the authors.

- There is a mention of stratified purposive sampling, it would be ideal to mention or depict the sampling frame in a figure (i.e. what were the stratas?).

- You mention the use of 6 local languages. Which are those languages? Were interpretors the ones running the interviews?

All the best with the revisions.

Reviewer #2: Review report

The authors have conducted a systematic and rigorous qualitative exploration of the barriers and facilitators of early intervention for childhood cataract. I have the following comments

1. They must provide a little more details about the Theoretical Determinants Framework in the background and introduction section. Has this framework been used in other public health settings? Has it been used in health care seeking behaviour settings?

2. There are some concerns in the methods section

a. What is the methodology adopted? This is not clear. Have the authors used grounded theory approach, narrative inquiry, phenomenology or case study? Have they taken an ethnographic approach? This methodological declaration is important and will help understand the rigor of the qualitative methodology.

b. It is important to have a reflexivity note in the methods section. In this note the researchers must position themselves in the research context and declare their biases, opinions, experiences and provide reflection on how these will impact on the research findings.

c. Most interviews have been conducted at the facility setting. There is a need to reflect on what kind of influence this choice of setting of the interviews will have on the findings.

d. Were any attempts made to triangulate the data obtained? There doesn’t seem to be any attempt at respondent triangulation, method triangulation. There is analytical triangulation reported, but other attempts at triangulation must also be presented.

e. It is not clear why the authors chose to use in depth interviews and not focus group discussions for data collection. This choice must be justified

f. How is the presence of a translator during the interviews likely to influence the interviews. This must be reflected on.

g. A hospital support staff was also present during the interviews. This must be reflected on. How will presence of a hospital staff during the interviews influence the findings. It is likely that some barriers related to hospital services may not be presented.

3. There are some issues in the analysis.

a. A strong positivist and quantitative approach has been used in what is a mainly interpretivist paradigm of research. The authors have attempted to count and quantify the frequency of various domains of the framework. They have concluded that this frequency across the interviews is likely to indicate its importance. This assumption has many problems. For example for this assumption to be true, the sampling must be representative and the sample size must meet the requirements of statistical power and confidence. But the sampling, sample size and other considerations are for qualitative analysis, which does not take quantification into consideration. Therefore applying estimates and generalizing them as indicating importance of a particular dimension is problematic. A priori declaration of the ontological and epistemological position of the paper will help resolve this issue. This paper works on a realist ontological position with a subjectivist epistemological approach. However, the analysis violates this position. I strong recommend removing quantification and reporting of estimates of importance of various dimensions.

b. The inter-rater agreement statistics is also problematic. Such quantifications are beyond the scope of such a data.

c. The authors have used a pre-existing framework and have looked for these themes in the data. This is largely an deductive approach. There is no justification for why they did not do a inductive method of theme building.

4. In the results section, the authors should provide richer descriptions of each theme with more elaborate verbatim quotes.

5. In the discussion section the authors have reported that they did not have any new themes that emerged from the data other than the framework that they have used. This is not possible. There is always scope for new themes to emerge from qualitative data.

6. PLOS authors have the option to publish the peer review history of their article (what does this mean?). If published, this will include your full peer review and any attached files.

Reviewer #1: **Yes: **Sikhumbuzo A Mabunda

Reviewer #2: **Yes: **Vijayaprasad Gopichandran

---

## [Author Response · Author response to Decision Letter 0]

17 Nov 2021

Dear Editor

Thanks for your valuable feedback to further enhance the quality of our manuscript. We have made a sincere attempt to address all the queries and comments raised by the reviewers and the details are given below in a tabular form for your easy reference. As suggested, we have also attached a revised manuscript with track changes and one without the changes highlighted. 

As stated earlier, none of the authors have any financial interest nor competing interest in this study. 

We are happy to furnish any information you may further require, and we are very much looking forward to hearing from you. 

Regards

Sheela

---

## [Decision Letter · Decision Letter 1]

22 Nov 2021

PONE-D-21-05307R1Barriers and enablers to access childhood cataract services across India. A qualitative study using the Theoretical Domains Framework (TDF) of behaviour changePLOS ONE

Dear Dr. Sethu,

Thank you for submitting your manuscript to PLOS ONE. After careful consideration, we feel that it has merit but does not fully meet PLOS ONE’s publication criteria as it currently stands. Therefore, we invite you to submit a revised version of the manuscript that addresses the points raised during the review process.

We look forward to receiving your revised manuscript.

Kind regards,

Leeberk Raja Inbaraj, MD

Academic Editor

PLOS ONE

Journal Requirements:

Additional Editor Comments (if provided):

I appreciate the authors for doing such wonderful work.Unfortunately, the revised version  does not fully meet the expectation of one of our reviewers.  I would suggest the authors to carefully go through the comments and address them meticulously. I am not in a position to accept the manuscript unless all the comments are satisfactorily addressed. I look forward for the revised version. 

Reviewers' comments:

Reviewer's Responses to Questions

**Comments to the Author**

1. If the authors have adequately addressed your comments raised in a previous round of review and you feel that this manuscript is now acceptable for publication, you may indicate that here to bypass the “Comments to the Author” section, enter your conflict of interest statement in the “Confidential to Editor” section, and submit your "Accept" recommendation.

Reviewer #1: All comments have been addressed

Reviewer #2: All comments have been addressed

2. Is the manuscript technically sound, and do the data support the conclusions?

Reviewer #1: Yes

Reviewer #2: Partly

3. Has the statistical analysis been performed appropriately and rigorously? 

Reviewer #1: Yes

Reviewer #2: N/A

4. Have the authors made all data underlying the findings in their manuscript fully available?

Reviewer #1: Yes

Reviewer #2: Yes

5. Is the manuscript presented in an intelligible fashion and written in standard English?

Reviewer #1: Yes

Reviewer #2: Yes

6. Review Comments to the Author

Reviewer #1: Congratulations to the authors on accomplishing this research. I am looking forward to reading it and more of their research.

Reviewer #2: Thank you for revising the manuscript and responding to the comments in detail. I am not convinced with the response to the questions related to analysis. Firstly the research paradigm used in this study is a social constructivist paradigm of understanding experiences, barriers, facilitators. The epistemological approach is subjective, using qualitative interviews. However the analysis method presents counting and quantification, which is a technique used in positivist approaches with objective epistemology. This is a conflict between the paradigm of research and analytical approach. The response that you have given is evidence of how this kind of quantification of the TDF framework is used in previous studies. Please provide a theoretical description of how such a positivist analytical approach is compatible with the social constructivist paradigm of research. It is for the same reasons that I also object to an inter-rater reliability. Inter rater reliability is a statistical method that follows a positivist paradigm. As this issue puts the core issue of validity of the findings of the paper into question, I would require a more detailed and theoretical response to this question.

7. PLOS authors have the option to publish the peer review history of their article (what does this mean?). If published, this will include your full peer review and any attached files.

Reviewer #1: **Yes: **Sikhumbuzo A. Mabunda

Reviewer #2: **Yes: **Vijayaprasad Gopichandran

---

## [Author Response · Author response to Decision Letter 1]

29 Nov 2021

We have provided a response as an attachment addressing the query raised by the reviewer.

---

## [Decision Letter · Decision Letter 2]

1 Dec 2021

Barriers and enablers to access childhood cataract services across India. A qualitative study using the Theoretical Domains Framework (TDF) of behaviour change

PONE-D-21-05307R2

Dear Dr. Sethu,

We’re pleased to inform you that your manuscript has been judged scientifically suitable for publication and will be formally accepted for publication once it meets all outstanding technical requirements.

Kind regards,

Leeberk Raja Inbaraj, MD

Academic Editor

PLOS ONE

Additional Editor Comments (optional):

Reviewers' comments:

Reviewer's Responses to Questions

**Comments to the Author**

1. If the authors have adequately addressed your comments raised in a previous round of review and you feel that this manuscript is now acceptable for publication, you may indicate that here to bypass the “Comments to the Author” section, enter your conflict of interest statement in the “Confidential to Editor” section, and submit your "Accept" recommendation.

Reviewer #2: All comments have been addressed

2. Is the manuscript technically sound, and do the data support the conclusions?

Reviewer #2: Yes

3. Has the statistical analysis been performed appropriately and rigorously? 

Reviewer #2: N/A

4. Have the authors made all data underlying the findings in their manuscript fully available?

Reviewer #2: Yes

5. Is the manuscript presented in an intelligible fashion and written in standard English?

Reviewer #2: Yes

6. Review Comments to the Author

Reviewer #2: Thanks for your response and revisions. The current version of the manuscript is acceptable for publication.

7. PLOS authors have the option to publish the peer review history of their article (what does this mean?). If published, this will include your full peer review and any attached files.

Reviewer #2: **Yes: **Vijayaprasad Gopichandran

---

## [Editor Report · Acceptance letter]

9 Dec 2021

PONE-D-21-05307R2 

Barriers and enablers to access childhood cataract services across India. A qualitative study using the Theoretical Domains Framework (TDF) of behaviour change 

Dear Dr. Sethu:

I'm pleased to inform you that your manuscript has been deemed suitable for publication in PLOS ONE. Congratulations! Your manuscript is now with our production department. 

Kind regards, 

on behalf of

Dr. Leeberk Raja Inbaraj 

Academic Editor

PLOS ONE